# Post-Translational Regulation of ARF: Perspective in Cancer

**DOI:** 10.3390/biom10081143

**Published:** 2020-08-04

**Authors:** Jinho Seo, Daehyeon Seong, Seung Ri Lee, Doo-Byoung Oh, Jaewhan Song

**Affiliations:** 1Environmental Disease Research Center, Korea Research Institute of Bioscience and Biotechnology (KRIBB), Daejeon 34141, Korea; sjh0130@kribb.re.kr (J.S.); dboh@kribb.re.kr (D.-B.O.); 2Department of Biochemistry, College of Life science and Biotechnology, Yonsei University, Seoul 03722, Korea; sdh0307@yonsei.ac.kr (D.S.); seungri2821@yonsei.ac.kr (S.R.L.); 3Department of Biosystems and Bioengineering, KRIBB School of Biotechnology, University of Science and Technology (UST), Daejeon 34113, Korea

**Keywords:** ARF, post-translational modification, transcriptional regulation, tumor suppressor, cancer, p14, ubiquitination, phosphorylation

## Abstract

Tumorigenesis can be induced by various stresses that cause aberrant DNA mutations and unhindered cell proliferation. Under such conditions, normal cells autonomously induce defense mechanisms, thereby stimulating tumor suppressor activation. ARF, encoded by the *CDKN2a* locus, is one of the most frequently mutated or deleted tumor suppressors in human cancer. The safeguard roles of ARF in tumorigenesis are mainly mediated via the MDM2-p53 axis, which plays a prominent role in tumor suppression. Under normal conditions, low p53 expression is stringently regulated by its target gene, MDM2 E3 ligase, which induces p53 degradation in a ubiquitin-proteasome-dependent manner. Oncogenic signals induced by MYC, RAS, and E2Fs trap MDM2 in the inhibited state by inducing ARF expression as a safeguard measure, thereby activating the tumor-suppressive function of p53. In addition to the MDM2-p53 axis, ARF can also interact with diverse proteins and regulate various cellular functions, such as cellular senescence, apoptosis, and anoikis, in a p53-independent manner. As the evidence indicating ARF as a key tumor suppressor has been accumulated, there is growing evidence that ARF is sophisticatedly fine-tuned by the diverse factors through transcriptional and post-translational regulatory mechanisms. In this review, we mainly focused on how cancer cells employ transcriptional and post-translational regulatory mechanisms to manipulate ARF activities to circumvent the tumor-suppressive function of ARF. We further discussed the clinical implications of ARF in human cancer.

## 1. Introduction

The cyclin-dependent kinase inhibitor 2A (*CDKN2a*) locus—frequently mutated or deleted in human cancer—encodes two different tumor suppressors, INK4a (referred to as p16^INK4a^) and ARF (referred to as p14^ARF^ in humans and p19^ARF^ in mice) [1,2]. These two proteins are translated from two different transcripts, α-transcript and β-transcript, respectively, which share sequences in exon 2 and 3. Although the INK4a and ARF transcripts share these sequences, both tumor suppressors display different amino acid sequences with distinctive functions due to the alternative reading frame and different transcription induction sites [3,4] (Figure 1).

Normal cells usually protect themselves from oncogenic signals, such as those induced by RAS and MYC signaling, by expressing tumor suppressors, such as INK4a and ARF, which prevent abnormal cellular growth that leads to tumor formation [5,6]. INK4a inhibits CDK4 and CDK6 activities required for G1 to S progression in the cell cycle by directly interacting with CDK4 and CDK6. This, in turn, interrupts the assembly of D-type cyclin and the CDK4/6 complex or inhibits the kinase activity of the cyclin D-CDK4/6 complex [7,8]. Due to the inhibition of CDK4/6, which phosphorylate RB, the hypo-phosphorylated form of RB predominates and interferes with the transcriptional activities of the E2F transcription factor 2 (E2F2) family members via direct binding, ultimately leading to cell cycle arrest [7] (Figure 1).

The major function of ARF is to stabilize and activate p53, resulting in cellular senescence or apoptosis. Mechanistically, ARF sequesters MDM2 (HDM2 in humans) via direct interaction, thereby blocking direct physical interaction of MDM2 with p53 in the cytosol and nucleus. As MDM2 is the E3 ligase for p53, ARF-induced MDM2 sequestration and inhibition prevents ubiquitination and proteasome-dependent degradation of p53, resulting in p53 stabilization [9,10]. p53 accumulation induces the formation of p53 tetrameric complexes in the nucleus, thereby activating the expression of genes related to cell cycle arrest or apoptosis (Figure 1). ARF involvement in MDM2 sequestration and the p53 axis has been considered a major part of the tumor-suppressive function of ARF; however, p53-independent roles of ARF are also emerging. The interactions of ARF with various proteins associated with cell proliferation and protection from oncogenic signals have been identified. For example, ARF inhibits E2F transcriptional activity by binding to E2F, thereby inducing cell cycle arrest [11,12,13,14]. The mitochondrial protein p32 enables ARF translocation to the mitochondria by interacting with the C-terminus of ARF, thereby promoting apoptosis [15]. Moreover, in response to oncogenic signaling, ARF interacts with various transcription factors, including MYC, nuclear factor-κB, and hypoxia-inducible factor 1-alpha, involved in various signaling pathways, thereby regulating cellular proliferation [16,17,18,19]. Recently, ARF has been shown to inhibit the transcriptional activity of nuclear factor E2-related factor 2, which protects cells from oxidative stress by activating the antioxidant program by binding to ARF, resulting in increased oxidative stress-induced ferroptosis [20]. Additionally, small mitochondrial ARF, p15^smArf^, has been identified as one of the ARF variants that is predominantly expressed in the mitochondria [21]. Although p15^smArf^, an N-terminal truncated form of full-length ARF, cannot activate p53, it could correct abnormal focal phenotype and spermatogenesis defects in ARF-null mice [22]. Structurally, ARF is composed of a high portion of basic and hydrophobic amino acids, allowing the aggregation of its recombinant form. This hampers the efforts made by many researchers to reveal its tertiary structure, further making its investigation difficult [23,24,25].

However, based on these observations, it was deduced that ARF formed a complex with numerous proteins to neutralize its charge and stabilize its spatial conformation. The importance of ARF in protection from aberrant tumor cell development has been further understood due to recent knowledge regarding the transcriptional and post-translational regulation of ARF by the newly revealed ARF regulators. This review focused on the newly extended regulatory network of ARF and its therapeutic implications in cancer.

## 2. Transcriptional Regulation of ARF

The paradoxical nature of oncogene-induced ARF expression has drawn attention to the search for ARF transcriptional factors (Table 1). While low ARF expression is maintained under normal conditions, significantly increased ARF expression is induced by oncogenic signals or DNA damage under stress conditions, subsequently prompting the activation of the fail-safe program. Fine-tuning of ARF expression by various transcription factors indicates that the tumor-suppressive activity of ARF can be manipulated under various stresses, depending on the nature of the stress signals (Figure 2). Here, we described several key transcription factors in the regulation of ARF mRNA expression.

### 2.1. Activators of ARF Transcription

MYC is an oncoprotein that regulates cell proliferation, differentiation, inflammation, and metabolism by regulating the transcription of various target genes [43]. To avoid unwanted effects resulting from its deregulation, MYC also induces ARF transcription, in turn initiating oncogene-induced senescence (OIS) [6]. OIS mechanisms can be either p53-dependent or -independent. As previously explained, ARF stabilizes p53 by trapping MDM2, inducing senescence or apoptosis [9,10]. Further, MYC expression also increased ARF mRNA in *p53*-null and wild type mouse embryonic fibroblasts (MEFs), suggesting that ARF might have p53-independent roles. Indeed, *Arf*-null cells showed resistance to MYC-induced apoptosis, indicating that ARF induction by MYC expression could be an important checkpoint in preventing aberrant cell proliferation through the activation of the fail-safe program [6]. As E2F is regulated by MYC, it has been identified as a regulator of MYC-mediated ARF expression [44,45]. The E2F family consists of five distinct E2F members. In particular, E2F transcription factor 1 (E2F1) and E2F2 activate ARF transcription. The conserved sequences of E2F1 and E2F2 are located in the ARF promoter, and activate ARF transcription in an RB-independent manner [44]. The E2F1-binding site on the ARF promoter (-231 to -205) is known as the E2F-responsive element [46]. On the contrary, another E2F member, E2F3b, functions as a negative regulator of ARF expression [47]. E2F3b deletion blocks mitogen-induced cell cycle entry and suppresses E2F-responsive gene expression, thereby inhibiting cell proliferation. The ARF promoter contains the E2F3b-specific binding site, which in response to E2F3b binding represses (rather than activating) ARF transcription. ARF stabilization, along with suppression of E2F-responsive genes, occurs in *E2F3b*-deficient cells, leading to p53 activation and triggering of G0/G1 cell cycle arrest [45].

DMP1, also referred to as cyclin D-binding Myb-like protein, can also induce cell growth arrest by inducing ARF expression [31,48]. Three DMP1 isoforms, DMP1α, β, and γ, are produced via alternative splicing, and they show differential functions in cell cycle progression [49]. DMP1α can bind to the consensus sequence of the ARF promoter (−189 to −181 in the transcription start site), thereby increasing ARF transcription [31]. DMP1α arrests the cell cycle in G0/G1 by increasing ARF expression. *Dmp1*-null MEFs show a delay in ARF accumulation when cells are sub-cultured and grow faster than wild type MEFs. H-Ras^V12^ expression results in tumorigenic phenotypes, and the delay in replicative senescence of *Dmp1*-null MEFs have been observed, suggesting that DMP1 functions as a tumor suppressor by increasing ARF expression [49]. In contrast to DMP1α, DMP1β inhibits ARF transcription [32]. DMP1β, which does not contain the DNA binding site of DMP1α, sequesters DMP1α, thereby inhibiting its binding to ARF promoter. DMP1β dose-dependently inhibits DMP1α-induced ARF promoter activity, implying that DMP1β antagonizes DMP1α transcriptional function and disturbs the interaction between DMP1α and ARF promoter [48]. In breast cancer patients, DMP1β is more expressed than DMP1α, suggesting the distinct role of DMP1β as an oncogenic factor rather than as a tumor suppressor [32].

FoxO is a subtype of the forkhead-box transcription factor, and has been identified as an ARF transcription factor [28]. Lymphomas, which express a dominant-negative FoxO mutant (dnFoxO), exhibit low ARF mRNA levels, regardless of the p53 status, implying that FoxO is involved in ARF transcriptional regulation. FoxO specifically binds to the FoxO-binding site in the first intron of ARF, thereby activating ARF transcription. FoxO inhibition in dnFoxO-expressing MYC-driven lymphomas promotes cell proliferation, while that in ARF-deficient MYC-driven lymphomas does not affect tumorigenicity, suggesting that FoxO inhibits MYC-derived lymphomagenesis by binding to the ARF promoter and inducing ARF expression.

Although ARF transcriptional regulation under conditions of oncogenic stresses has drawn significant attention, ARF regulation is an important process in embryogenesis [50,51,52]. Similar to *Arf*-deficient embryos, *Tgfβ2*-deficient embryos showed eyes with a hyperplasia phenotype and low ARF expression and acidic β-galactosidase on embryonic day 13.5, implying that TGFβ2 might function as a positive regulator of ARF transcription during embryogenesis [29]. Smad 2 and 3 appear to bind to the proximal region of the *ARF* locus in a TGFβ2-dependent manner, and inducing histone acetylation and RNA pol II recruitment. This in turn promotes ARF promoter remodeling and activation [30]. Unlike TGFβ2, TGFβ1 participates in a signaling pathway that negatively regulates ARF transcription in B-cell lymphoma [41]. In mutant p53-expressing B-cell lymphoma cells, ARF showed tumorigenic effects, different from its original tumor-suppressive function, by stabilizing mutant p53 through MDM2 inhibition. In this case, TGFβ1 activation in the B-cell lymphoma resulted in decreased E2F1 expression, leading to reduced ARF transcription. Low ARF expression leads to destabilization of mutant p53, thereby inducing cell cycle arrest.

### 2.2. Suppressors of ARF Transcription

The polycomb group transcription factor, BMI-1, was identified as an oncogene that cooperates with MYC in mouse lymphomas [53,54,55,56]. *Bmi-1*-deficient cells showed cellular senescence phenotypes, including decreased cell proliferation, defective S-phase cell cycle, cytoplasmic enlargement, unresponsiveness to growth factors, and increased acidic β-galactosidase, indicating that *Bmi-1* deficiency was closely related to cellular senescence [57]. While there was no change in p21, p27, and p53 expression in *Bmi-1*-deficient cells, ARF and INK4a were upregulated. In the *Bmi-1*-deficient mouse model, a significantly small size and an ataxia phenotype were observed, and these phenotypes could be restored by intercrossing the BMI-1-deficient mice with *Ink4a/Arf* locus-deficient mice [57].

Another polycomb group protein, CBX7, has been identified as a transcriptional repressor of ARF [58]. CBX7 expression prevents ARF and INK4a mRNA accumulation, thereby delaying replicative senescence. When cells were sub-cultured, CBX7 levels gradually decreased, and thus, CBX7 expression was completely lost in the senescent cells. Furthermore, *Cbx7*-deficient cells showed severe growth arrest and high ARF and INK4a levels, thereby indicating that CBX7 and senescence suppressed each other. CBX7-induced suppression of ARF and INK4a expression occurs in a BMI-1-independent manner, suggesting that CBX7 may repress ARF and INK4a transcription by interacting with another polycomb-repressive complex-1 (PRC1) subset, but not with BMI-1.

Polycomb-repressive complex-2 (PRC2) represses ARF transcription by associating with Twist-1 [42]. Twist-1 expression in bone marrow-derived mesenchymal stem/stromal cells (BMSCs) results in increased proliferation and decreased β-galactosidase-positive cell counts. ARF expression is reduced by Twist-1, implying that Twist-1 may function as a negative regulator of ARF and prevent cellular senescence. Twist-1 recognizes H3K27me3—which interacts with Ezh2, a PRC2 component—on the ARF locus, subsequently recruiting PRC2, which suppresses ARF transcription. Additionally, Twist-1 suppresses the expression of E47, which is an INK4a transcriptional activator. Collectively, Twist-1 suppresses ARF and INK4a expression by recruiting PRC2 and decreasing E47 expression, thereby functioning as a negative regulator of cellular senescence.

Through senescence bypass library screening, TBX2 has been identified as a transcriptional suppressor of ARF [37]. TBX2 prevents BMI-1 deletion-induced premature senescence. The accumulation of ARF in *Bmi-1*-deficient cells was reduced upon TBX2 expression without any change in INK4a, suggesting that TBX2 specifically inhibits ARF expression.

EGFR, a membrane-bound receptor tyrosine kinase, serves as a mediator of proliferative signaling by activating diverse downstream components [59]. Suppression of ARF expression has recently been identified as being among the EGFR oncogenic mechanisms [34]. Upon binding to the epidermal growth factor, EGFR activates and interacts with the catalytic subunit type 3 (VPS34) of phosphatidylinositol 3-kinase, while translocating from the plasma membrane to the nucleus. The EGFR-VPS34 complex inhibits ARF transcription by binding to the AT-rich sequence in the ARF promoter, thereby suppressing the ARF-mediated fail-safe program [35].

## 3. Post-Translational Regulation of ARF Regulates Its Functional Roles in Cellular Physiology

The control of ARF expression at the transcriptional level has been sufficiently described; accumulating evidence regarding the post-translational regulation of ARF suggests that oncogenic stress-mediated ARF induction may be associated with molecular mechanisms that regulate protein levels and functions. In particular, the involvement of phosphorylation, ubiquitination, and proteasome- and lysosome-dependent pathways in the post-translational regulation of ARF has recently been observed.

### 3.1. ARF Phosphorylation

Phosphorylation is one of the well-known post-translational modifications involved in numerous cellular signaling pathways. Serine, threonine, or tyrosine residues of proteins are commonly phosphorylated by kinases, which are enzymes that add the phosphate group onto the target amino acids via esterification reactions [60]. Phosphorylation at these amino acids leads to conformational changes in proteins as a result of the change in charge of proteins, thereby inducing protein activation/deactivation or promoting protein degradation in a proteasome-dependent manner. Additionally, phosphorylation can change the affinity in protein–protein interactions (PPIs); thus, in diverse signaling pathways, phosphorylation cascades function as key events in regulating cellular signaling.

Despite the important roles of phosphorylation in protein activity regulation, only one phosphorylation site has been identified in ARF (Figure 3). Inoue et al. (2005) found that 12-o-tetradecanoyl-phorbol 13-acetate (TPA)-mediated protein kinase C alpha (PKCα) activation stabilized the ARF protein [61]. TPA treatment stabilizes ARF by activating PKCα, but not members of other PKC families. Here, binding of PKCα to ARF in the cytoplasmic compartment induces ARF phosphorylation at threonine 8, thereby resulting in the stabilization of ARF [62]. The phospho-mimetic mutant of ARF (T8D) showed a longer half-life and lesser nucleolar localization than wild type ARF. Additionally, T8D was no longer involved in cell growth retardation and could not affect the MDM2/p53 axis, suggesting that ARF phosphorylation inhibited ARF tumor-suppressive function. Recently, Fontana et al. (2018) reported that ARF phosphorylation at threonine 8 by PKCα was involved in the regulation of the focal adhesion kinase (FAK) pathway. ARF phosphorylation in the cytoplasm promotes cell spreading and alleviates anoikis [63]. Collectively, ARF phosphorylation appears to function as a cell survival factor rather than a tumor suppressor. Further molecular studies on the roles of ARF phosphorylation induced by PKCα or other kinases are required.

### 3.2. ARF Regulation via Degradation

#### 3.2.1. The Ubiquitin-Proteasome System (UPS)

Ubiquitination is an important cellular process in which small molecules called ubiquitins are added to the target proteins; it occurs in three steps, requiring three enzymes, i.e., E1 ubiquitin-activating enzymes, E2 ubiquitin-conjugating enzymes, and E3 ubiquitin ligases. Ubiquitin forms a thioester bond with E1 in an ATP-dependent manner, resulting in the initial activation of ubiquitin. Then, ubiquitin is transferred to E2. E2-conjugated ubiquitins are transferred to the target proteins by E3 ligases; this is the rate-determining step of ubiquitination [64]. More than 1000 E3 ligases have been identified in humans, implying that ubiquitination regulates numerous target proteins and cell signaling pathways, and by extension, cellular physiology. The carboxyl-terminal residue of ubiquitins (G76) is linked to the lysine residue of target proteins via covalent bonds, leading to target protein activation or a change in localization. Additionally, ubiquitins can form polymers, known as polyubiquitin chains; this process is known as polyubiquitination [65]. Polyubiquitin chains are generated by adding ubiquitins to one of seven lysines (K6, 11, 27, 29, 33, 48, or 63) or methionine 1 (M1). Each polyubiquitin chain specifically regulates protein functions. In particular, the K48-linked polyubiquitin chain promotes target protein degradation in a proteasome-dependent manner, and the K63- or M1-linked polyubiquitin chains provide a platform for signaling transduction.

As human ARF does not contain any lysine residue, studies on ARF ubiquitination mainly focused on exploring the possibility of N-terminal ubiquitination of ARF [66]. In humans and mice, ARF turnover is inhibited upon treatment with MG132, a proteasome inhibitor, suggesting that ARF protein is continuously degraded in a proteasome-dependent manner. Overexpression of ubiquitin and human ARF or lysine-deficient mouse ARF (K26R) induces ARF ubiquitination, resulting in the accumulation of ubiquitinated ARF upon MG132 treatment. The N-terminal mutant ARF, which is recalcitrant to processing by methionine aminopeptidase, has fewer ubiquitinated forms and a longer turnover time than wild type ARF, thereby implying that N-terminal ubiquitination of ARF by an unknown E3 ligase destabilizes ARF in a proteasome-dependent manner.

ULF, also referred to as thyroid hormone receptor interactor 12, has been identified as the first E3 ligase for ARF [67]. ULF depletion stabilizes p53 by increasing ARF protein levels without altering those of ARF mRNA, leading to cell cycle arrest. ARF ubiquitination by ULF increases when NPM, which binds to ARF and arrests it in the nucleolus, is inhibited, implying that ULF-mediated ARF ubiquitination occurs in the nucleolus. Therefore, NPM appears to protect ARF by sequestering it away from the E3 ULF in the nucleus. Recently, several proteins have been reported as regulators of ULF-mediated ARF degradation. MYC binds to ULF and then inhibits ARF-ULF interaction [67,68]. The inhibitory function of MYC in ULF-mediated ARF ubiquitination does not depend on MYC transcriptional activity, suggesting that MYC can control ARF levels via transcriptional as well as post-translational regulation. Tumor necrosis factor receptor (TNFR)-associated death domain (TRADD) independently inhibits ULF-mediated ARF ubiquitination and stabilizes ARF proteins via the TNFR signaling pathway [69]. TRADD actively shuttles between the cytoplasm and nucleus. Nuclear TRADD interacts with ULF and abrogates ARF-ULF interaction. Upon expression of H-Ras^V12^ in cells, increased TRADD expression, followed by ARF stabilization is observed, resulting in the promotion of OIS. Nucleostemin (NS) is a GTPase that is localized in the nucleolus and is involved in overall protein synthesis [70]. NS stabilizes ARF in two ways. The first involves the solidification of the NPM-ARF complex in the nucleolus. NS depletion leads to reduced NPM-ARF interaction, subsequently resulting in ARF destabilization. The second involves the inhibition of ULF-ARF interaction. NS binds to ULF and abolishes ULF-mediated ARF ubiquitination. Glioma tumor-suppressor candidate region gene 2 (GLTSCR2), which is a nucleolar protein, has also been identified as a modulator of ARF nucleoplasmic localization and stability [71]. GLTSCR2 can interact with ARF, leading to ARF nucleoplasmic localization. This increases ULF-ARF interaction, thereby inducing ARF degradation in a proteasome-dependent manner.

MKRN1 is an E3 ligase that regulates diverse cellular signaling pathways by leading to the ubiquitin-mediated proteasome-dependent degradation of target proteins. Ko et al. (2012) reported that MKRN1 ubiquitinates ARF, thereby leading to the proteasome-dependent degradation of ARF [72]. MKRN1 depletion promotes cellular senescence in gastric cancer cell lines, which can be rescued by ARF co-depletion. MKRN1 overexpression decreases ARF protein levels and half-life, and this can be blocked by MG132 treatment. A change in ARF localization from the nucleolus to the nucleus and cytoplasm can be observed upon the induction of MKRN1 expression, indicating that MKRN1-mediated ARF ubiquitination—followed by proteasomal degradation—occurs in the nucleus and cytoplasm. Another E3 ligase, Siva1, has been identified as a direct E3 ligase for ARF [73]. Siva1 interacts with ARF and induces the export of ARF from the nucleolus. Siva1-ARF interaction increases K48-linked polyubiquitination of ARF, resulting in ARF destabilization via the proteasome pathway. Siva1 depletion leads to p53 stabilization through the ARF-MDM2 axis, resulting in cell cycle arrest at the G1 stage. Beta-transducin repeat-containing protein 2 (β-TrCP2), a phosphorylation-dependent E3 ligase, ubiquitinates and degrades mouse ARF (p19^ARF^) in response to serum stimulation, thereby promoting cell proliferation via growth factor signaling [74]. β-TrCP2 deletion in MEFs increased p19^ARF^ levels and suppressed cell proliferation. β-TrCP2-induced p19^ARF^ ubiquitination and degradation require phosphorylation at serine 75. This is performed by ribosomal protein S6 kinase beta-1 (S6K1), which is activated in response to growth factor- or nutrient-dependent mammalian target of rapamycin (mTOR) signaling. As only p19^ARF^—but not p14^ARF^ or other species—contains the β-TrCP2 and S6K1 degron, the regulation of p19^ARF^ via the mTOR-S6K1-β-TrCP2 axis was restricted in the mouse model.

Recently, deubiquitinating enzymes (DUBs), E3 ligase counter enzymes that remove ubiquitin molecules from the target proteins by cleaving the thioester bonds of ubiquitins, have been reported to regulate ARF. USP10 has been identified as the first direct DUB for ARF [75]. USP10 is transcriptionally activated during MYC expression-induced OIS. MYC activates USP10 transcription by binding to the second E-box sequence located upstream of the USP10 transcription start site. USP10 deubiquitinates ARF by interacting with the N-terminus of ARF, thereby stabilizing the ARF protein by blocking the proteasome-mediated ARF degradation. USP10 depletion or knockout alleviates MYC expression-induced OIS, suggesting that USP10 functions as a key factor in the oncogenic stress-induced fail-safe program. Additionally, ubiquitin-specific-processing protease 7 (USP7) regulates ARF levels [76]. USP7 depletion in hepatocellular carcinoma (HCC) cell lines decreases proliferation and increases the number of cells arrested at the G1 stage. USP7 forms a complex with ULF, and protects it from proteasome-mediated degradation by deubiquitination. ULF stabilization promotes ARF degradation, thereby promoting HCC cell proliferation.

#### 3.2.2. Chaperone-Mediated Autophagy (CMA)

Substrates containing the HSC70 canonical binding motif are recognized by a chaperone-co-chaperone complex, and are then bound by lysosome associated membrane glycoprotein 2 (LAMP2A) [77]. Interaction of the substrate-chaperone complex with LAMP2A results in LAMP2A multimerization and substrate internalization into the lysosome followed by substrate degradation in a process known as CMA [77].

Recently, Han et al. (2017) reported that HSP90, a molecular chaperone, regulated ARF turnover via CMA [78]. Treatment with geldanamycin, an inhibitor of HSP90, or HSP90 depletion by siRNA promotes cellular senescence, while ARF co-depletion rescues the increase in senescence. Interestingly, HSP90 binds to ARF in a CHIP-dependent manner. CHIP is a co-chaperone protein and an E3 ligase [79]. The formation of the HSP90, ARF, and CHIP complex allows ARF to interact with LAMP2A, which is subsequently degraded via the lysosomal degradation pathway. It must be noted that the E3 ligase activity of CHIP is not essential for HSP90-ARF-CHIP ternary complex formation and subsequent ARF degradation, suggesting that ubiquitination is not required.

### 3.3. Protein–Protein Interaction (PPI)

NPM (also called B23) is a multifunctional protein that controls a variety of cellular phenomena, including proliferation, genomic stability maintenance, cell death, rRNA processing, and ribosome biogenesis [80]. It is abundantly expressed in the nucleolus, and actively shuttles between the nucleus and cytoplasm. Over the past few years, the NPM-ARF interplays have been extensively studied. NPM enhances the nucleolar translocation of ARF by forming an NPM-ARF complex, thereby blocking ARF-MDM2 interaction—in the nucleus—and ARF-mediated p53 stabilization [81,82]. NPM depletion promotes ARF translocation from the nucleolus to the nucleus and cytoplasm, and subsequently leads to ARF destabilization. Furthermore, ARF was found to be localized in the cytoplasm of NPM-deficient cells or acute myeloid leukemia cells containing mutant NPM, suggesting that NPM played a central role in ARF nucleolar localization and stability [82,83]. Under genotoxic stress, the NPM-ARF complex translocates to the nucleus in a c-Jun-NH_2_-kinase (JNK) pathway-dependent manner [84]. c-Jun interacts with NPM in the nucleolus under normal conditions. When cells are exposed in ultraviolet radiation, JNK-induced phosphorylation of c-Jun at threonine 91 and 93 leads to the nuclear translocation of the c-Jun-NPM-ARF complex. Recently, another DNA damage response pathway—the ataxia telangiectasia mutated (ATM)-mediated pathway—has been identified as a novel signaling pathway that regulates ARF localization by modulating serine/threonine-protein kinase Nek 2 (Nek2)-dependent NPM phosphorylation [85]. Under normal conditions, Nek2 phosphorylates NPM at serine 70 and 88, thereby enhancing NPM-ARF interaction. When cells are treated with doxorubicin, ATM activates protein phosphatase-1 (PP1)—a counter phosphatase of Nek2—thereby, inducing NPM dephosphorylation, which causes nuclear localization of ARF by abrogating NPM-ARF interaction, and destabilizing ARF via ULF-mediated ubiquitination. Another kinase, protein kinase B (AKT), also regulates ARF stability and localization via NPM phosphorylation [86]. AKT interacts with NPM, resulting in NPM phosphorylation at serine 48 (located in the oligomerization domain). Structure analysis revealed that AKT-induced NPM phosphorylation at serine 48 is incompatible with NPM oligomerization due to steric clashes, thereby resulting in the nuclear and cytoplasmic localization of the NPM-ARF complex. The nuclear localization of the NPM-ARF complex inhibits MDM2 function and stabilizes p53. As AKT directly phosphorylates and activates MDM2, MDM2 inhibition by AKT-mediated NPM phosphorylation appears to occur in an ARF expression-dependent manner, but the detailed molecular mechanisms remain unclear.

In addition to NPM, several other ARF binding partners have been reported as regulators of ARF localization and stability. TBP-1 has been identified as an ARF stabilizer [87]. TBP-1, a proteasome component, forms a complex with ARF by binding to the 1-39 amino acids of ARF, which blocks proteasome-dependent ARF degradation, thereby resulting in ARF stabilization [88]. As the first two-three amino acids of ARF cannot influence TBP-1-mediated stabilization, TBP-1-mediated ARF stabilization appears to occur in an N-terminal ubiquitination-independent manner. Another component of the proteasome, REGγ, promotes ARF degradation in a ubiquitination-independent manner [89]. REGγ binds to ARF and directly induces its destabilization in a proteasome-dependent manner. Recently, MDM2 has been identified as a negative regulator of ARF [90]. MDM2 overexpression results in ARF destabilization via the ubiquitination-independent proteasome pathway. PKCα-induced ARF phosphorylation appears to block MDM2-mediated ARF degradation, indicating that MDM2 interacts with dephosphorylated ARF and transports it to the proteasome. Additionally, PANO, a nucleolar protein, interacts with ARF [91]. PANO overexpression increases p53 levels via ARF stabilization, thereby leading to increased apoptosis.

## 4. Post-Translational Regulation of ARF in Human Cancer

Over the last three decades, many mechanistic and clinical observations have indicated a close relationship between ARF and tumor progression in animal and human cancer models. *ARF*-null mice generated by specifically targeting exon 1β in the *CDKN2a* locus spontaneously develop numerous tumors, including sarcomas, lymphomas, and lung carcinomas, resulting in death within one year [92,93,94]. Treatment of *ARF*-knockout mice using carcinogens, such as dimethylbenz(a)anthracene (DMBA) or X-ray, promotes tumor progression, leading to a short life span of six months with drastic tumor development. Epigenetic modifications on ARF promoters have been identified in patients with a wide spectrum of tumors. As the ARF promoter contains a CpG island, ARF silencing by hypermethylation of these promoter sequences has frequently been reported in various human cancers, such as breast, bladder, colon, liver, gastric, lung, oral, prostate, and brain cancer [95,96,97,98,99,100,101,102,103,104,105,106,107,108,109,110]. Additionally, homologous deletion or loss of heterozygosity on the *CDKN2a* locus has also been frequently reported in numerous cancers, including breast, bladder, liver, lung, oral, prostate, and kidney cancer [97,102,106,107,108,109,110,111,112]. Furthermore, point mutations, including short deletion, insertion, and missense mutations, on the ARF exon 1β or exon 2 have been found in familial melanoma [113,114,115,116].

Genetic and epigenetic modifications in the *CDKN2a* locus have been well characterized in diverse cancers; the uncoupling of ARF mRNA and protein expression has also been observed in lung cancer [117,118]. Low ARF expression without deleterious mutations in the *CDKN2a* locus is found in non-small cell lung cancer (NSCLC), implying that post-translational regulation of ARF may be involved in cancer development (Table 2). In malignant gastric cancer patients, increased MKRN1 and low ARF expression is observed in well-differentiated adenocarcinoma, while low MKRN1 and high ARF expression is detected in poorly differentiated adenocarcinoma [72]. MKRN1 and ARF expression in gastric cancer patients shows an inverse correlation, suggesting that MKRN1-mediated ARF degradation may have an important clinical implication in gastric cancer. Additionally, a significant correlation between TRADD and ARF has been observed in invasive breast cancer [69]. TRADD expression does not correlate with estrogen, progesterone, and receptor tyrosine-protein kinase erbB-2 (HER2) receptor expression; however, it shows a positive correlation with ARF expression and relapse-free survival rates. In HCC patients, high USP7 and ULF expression, and low ARF expression, is observed [76]. Low USP7, ULF expression indicates high overall survival rates in HCC patients, suggesting that USP7-ULF-mediated ARF regulation is a useful parameter for predicting HCC prognosis. Various studies have been performed on the correlation between post-translational regulators and ARF in NSCLC. A panel of lung carcinomas with reduced ATM levels showed high ARF expression and low PP1 phosphorylation [85]. Furthermore, the inverse correlation between ATM and ARF was strengthened in a high INK4a expression panel, indicating that the association between ATM and ARF in NSCLC was mediated by post-translational regulation, not *CDKN2a* locus aberration. High HSP90 and CHIP expression, with reduced ARF expression, has also been identified in NSCLC [78]; high HSP90 or combined HSP90/CHIP expression, with low ARF expression, indicates worse overall survival rates. The inverse correlation between combined HSP90/CHIP expression and ARF is more substantial in advanced NSCLC than in the early stage, suggesting that combined HSP90/CHIP expression may be an independent prognostic marker for early detection of NSCLC. Recently, MYC, USP10, and ARF expression in NSCLC have been determined [75]. USP10 and ARF expression show a positive correlation, whereas MYC expression does not correlate with USP10 and ARF expression. MYC appears to be closely related to the disruption of the fail-safe program in NSCLC. Additionally, the dual loss of USP10 and ARF expression is frequently observed in small intestinal adenocarcinoma and ovarian cancer patients, thus implying that low combined USP10/ARF expression is a prognostic marker of small intestinal adenocarcinoma and ovarian cancer [119,120].

## 5. Conclusions and Perspectives

Since its discovery, ARF regulation has been extensively studied because of its importance in the determination of cell fate in response to oncogenic signals. ARF induces cellular senescence, cell cycle arrest, and apoptosis via p53-dependent or -independent pathways in response to oncogenic stress, which activates the fail-safe program [2,4]. When cells are abrogated in the ARF-induced fail-safe program due to deregulated ARF expression, cells fail to activate defense mechanisms in response to oncogenic stresses, such as DNA damage, oncogene activation, and oxidative stresses, leading to uncontrolled proliferation [2,4].

Numerous transcription factors have been identified as transcriptional regulators of ARF, and their clinical implications in cancer have been reported. However, the uncoupling of ARF mRNA and protein expression in human cancer remains unclear. For a decade, the molecular mechanisms underlying post-translational regulation of ARF have been suggested as key events in maintaining the balance of ARF expression in response to various signaling pathways. Most post-translational regulatory mechanisms control the nucleolar, nuclear, and cytoplasmic localizations of ARF, leading to stabilization/destabilization. In particular, PKCα-induced ARF phosphorylation leads to cytoplasmic localization and a change in the functional role of ARF (Figure 3). Phosphorylated ARF regulates cell spreading, rather than cell cycle arrest, via the FAK signaling pathway, suggesting that post-translational modification of ARF alters its function [63,121]. Recently, the oncogenic functions of ARF have been reported. ARF promotes tumorigenesis in prostate cancer by stabilizing Slug, interacts with metallopeptidase-7, and shows high protein expression in aggressive lymphoma, invasive bladder cancer, and thyroid cancer [122,123,124,125,126,127,128]. Furthermore, the fail-safe program-independent functions of ARF such as autophagic and oxidative stress-sensing functions have been continually accumulated. However, the mechanisms by which ARF switches from tumor-suppressive to oncogenic (or other) functions remain unclear [20,127,129,130,131]. Further studies on the post-translational regulation of ARF will provide a comprehensive understanding of these contradictory functions, thereby presenting a broad understanding of the ARF network in human cancer.

Many cancer patients show genetic and epigenetic modifications in the *CDKN2a* locus. However, these modifications do not explain why several cancer patients show low ARF expression without *CDKN2a* locus aberration. As described above, diverse factors are involved in the post-translational regulation of ARF, demonstrating meaningful clinical implications in cancer patients. Targeting the fine-tuned post-translational regulation of ARF would enable the development of independent prognostic markers and therapeutic strategies for various cancers.

## Figures and Tables

**Figure 1 biomolecules-10-01143-f001:**
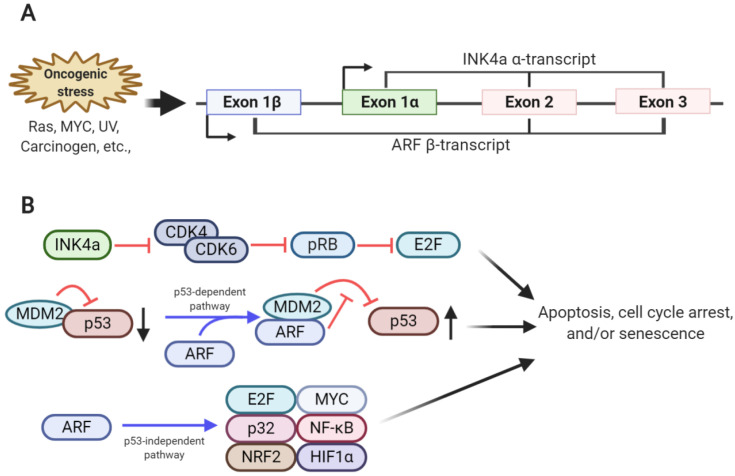
Overview of the genomic structure and function of the *CDKN2a* locus. (**A**) Two transcripts, α-transcript (encoding INK4a) and β-transcript (encoding ARF), are transcribed from the *CDKN2a* locus in response to oncogenic stresses. Although these two transcripts share exon 2 and 3 sequences, they have alternative reading frames, and thus are translated into two different proteins. (**B**) INK4a inhibits cyclin-dependent kinase 4/6 (CDK4/CDK6) activity, leading to an increase in hypo-phosphorylated retinoblastoma (RB) levels. Hypo-phosphorylated RB blocks E2F function, subsequently inducing cell cycle arrest. ARF binds to mouse double minute 2 homolog (MDM2), which accumulates in the nucleolus and inhibits E3 ligase activity. This leads to p53 stabilization, inducing cell cycle arrest and apoptosis. ARF induces apoptosis, cell cycle arrest, and senescence in a p53-independent manner.

**Figure 2 biomolecules-10-01143-f002:**
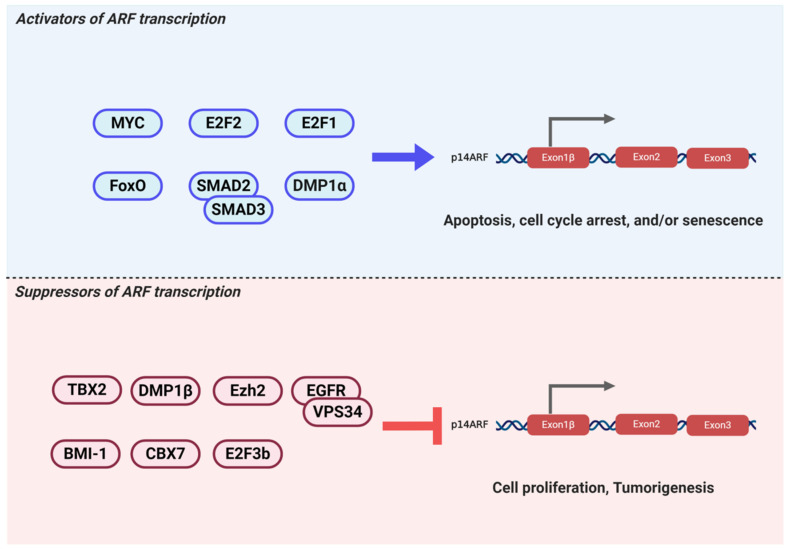
A number of transcriptional factors positively or negatively regulate ARF transcription. Smads, DMP1α, E2Fs, MYC, and FoxO activate ARF transcription. E2F3b, enhancer of zeste homolog 2 (Ezh2)/Twist-1, chromobox protein homolog 7 (CBX7), T-box transcription factor 2 (TBX2), B-cell-specific Moloney murine leukemia virus integration site 1 (BMI-1), and epidermal growth factor receptor (EGFR) directly binds to the ARF promoter and suppress ARF transcription. DMP1β interacts with DMP1α, subsequently blocking the binding of DMP1α to the ARF promoter. Furthermore, the transforming growth factor beta 1 (TGFβ1) signaling pathway negatively regulates ARF transcription by inhibiting E2F1 expression.

**Figure 3 biomolecules-10-01143-f003:**
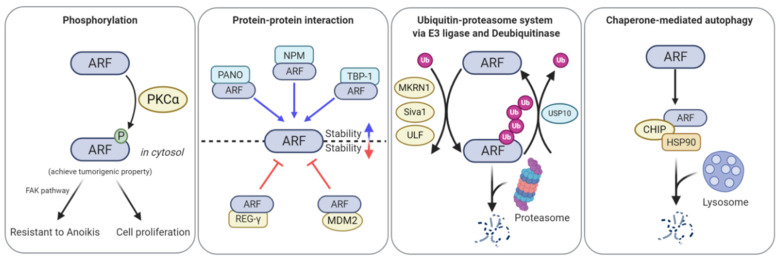
ARF function, stability, and localization are regulated by phosphorylation, ubiquitination, PPIs, and chaperone-mediated autophagy (CMA). PKCα-induced ARF phosphorylation leads to the cytoplasmic localization of ARF, thereby promoting cell spreading and alleviating anoikis. Makorin ring finger protein 1 (MKRN1), Siva1, and ubiquitin ligase for ARF 1 (ULF1) ubiquitinate ARF, leading to its proteasome-mediated degradation. Ubiquitin-specific peptidase 10 (USP10) directly binds to ARF and then detaches ubiquitin from ARF, thus stabilizing ARF. Proapoptotic nuclear protein 1 (PANO), nucleophosmin (NPM), and tat-binding protein-1 (TBP-1) can bind to ARF, thereby preventing its degradation. Proteasome activator complex subunit 3 (REG-γ) and MDM2 interact with ARF and then transport ARF to the proteasome, thereby resulting in ARF degradation. C-terminus heat shock cognate 71 kDa protein (HSC70)-interacting protein (CHIP)-heat shock protein 90 (HSP90) forms a complex with ARF, leading to degradation of ARF in a lysosome-dependent manner.

**Table 1 biomolecules-10-01143-t001:** The relationship between the transcription factors and ARF expression in human cancer.

Transcription Factor	Cancer Type	Correlation with ARF Expression	Molecular Mechanism	Ref.
MYC	Acute myeloid leukemia	Positive correlation with ARF The combined expression of high MYC and ARF in AML Patients with low ARF expression worsen overall survival rates	MYC overexpression increases ARF mRNA transcription. ARF null mice exhibit resistance to MYC-driven apoptosis.	[26]
E2F1/E2F2	Colon cancer	Positive correlation with ARF The combined expression of high E2Fs and ARF in colon cancer	E2Fs bind to the conserved sequence of ARF promoter, increasing ARF transcription. Overexpression of E2F1 leads to G2/M arrest with increase in ARF protein levels.	[27]
FoxO	Primary lymphoma	Positive correlation with ARF FoxO proteins have an instructive role in regulating ARF expression during MYC-induced lymphomagenesis	FoxO increases ARF transcription via interacting with FoxO-binding site region in the first intron of ARF. Lymphomas expressing a dominant-negative mutant of FoxO (dnFoxO) have low levels of ARF mRNA regardless of the p53 status.	[28]
TGF-β2/ SMAD2/3	Unknown	Positive correlation with ARF TGFβ2-deficient embryos show hyperplasia phenotype in the eyes at embryonic day 13.5 with low ARF expression	SMAD2/3 bind to a proximal region of the ARF locus in a TGFβ2-dependent manner.	[29,30]
DMP1α	Unknown	Positive correlation with ARF	DMP1α binds to the consensus sequence of the ARF promoter, leading to an increase in ARF transcription.	[31]
DMP1β	Breast cancer	Inverse correlation with ARF The correlation between high DMP1β expression and shorter survival of breast cancer patients	DMP1β binds to DMP1α, which inhibit its transcriptional activity, thereby leading to a decrease in ARF transcription. High DMP1β and low DMP1α expression due to alternative splicing is frequently observed in breast cancer patients.	[32,33]
EGFR/VPS34	Lung cancer	Inverse correlation with ARF The expression of low ARF in lung tumors harbouring constitutive active mutant EGFR	Active EGFR interacts with VPS34, which moves to the nucleus, thus inhibiting ARF expression via binding to the AT-rich sequence of the ARF promoter.	[34,35]
E2F3b	Hepatocarcinoma	Inverse correlation with ARF The expression of high E2F3 in hepatocellular carcinoma (HCC)	E2F3b represses ARF mRNA expression via binding to ARF promoter. E2F3b induces G1/S phase transition and markedly increases cell proliferation, but has a minor effect on apoptosis.	[36]
TBX2	Breast cancer	Inverse correlation with ARF TBX2 amplification in human breast cancer	ARF expression in BMI-1 deficient cells is suppressed by TBX2 without any change in INK4a level.	[37]
BMI-1	Breast cancer	Inverse correlation with ARF	Overexpression of BMI-1 results in the elevation of expression of polycomb group (PcG)-target genes followed by the inhibition of ARF expression.	[38]
	Prostate cancer	Inverse correlation with ARF The combined expression of high BMI-1 and low ARF in prostate cancer	BMI-1-expressing DU145 cells form drastic large tumors in NOD/SCID mice.	[39]
CBX7	Prostate cancer	Inverse correlation with ARF	CBX7 ablation retards cell proliferation via the ARF/p53 and INK4a/Rb pathways.	[40]
TGF-β1	B-cell lymphoma	Inverse correlation with ARF	In B-cell lymphoma expressing mutant p53, activation of TGFβ1 leads to a decrease in E2F1 expression, leading to the reduction in ARF transcription. The low expression of ARF induces the destabilization of mutant p53.	[41]
Twist/Ezh2	Unknown	Inverse correlation with ARF	Twist-1 recognizes H3K27me3 on the ARF locus followed by interaction with Ezh2, which leads to suppression of ARF transcription via PRC2 complex.	[42]

(Related to ‘2. Transcriptional regulation of ARF’).

**Table 2 biomolecules-10-01143-t002:** The relationship between the ARF post-translational regulator and ARF expression in human cancer.

Post-Translational Regulator	Cancer Type	Correlation with ARF Expression	Molecular Mechanism	Ref.
MKRN1	Gastric adenocarcinoma	Inverse correlation with ARF The combined expression of high MKRN1 and low ARF in well-differentiated adenocarcinoma	MKRN1 promotes ARF ubiquitination, which leads to the proteasome-dependent degradation of ARF	[72]
TRADD	Invasive breast cancer	Positive correlation with ARF Low TRADD expression correlates with poor prognosis.	TRADD competes with ULF for interaction with ARF, protecting ARF from ULF-mediated ubiquitination.	[69]
ATM	Lung carcinoma	Inverse correlation with ARF	ATM-PP1 axis inhibits Nek2 kinase activity, which induces the de-phosphorylation of NPM, thus leading to the nucleoplasm localization and degradation of ARF.	[85]
USP7/ULF	Hepatocarcinoma	Inverse correlation with ARF The combined expression of low USP7 and ULF worsen overall survival rates.	USP7 forms a complex with ULF that protects ULF protein from proteasome-mediated degradation via removal of ubiquitin.	[76]
HSP90/CHIP	NSCLC	Inverse correlation with ARF The combined expression of high HSP90, CHIP, and low ARF worsen overall survival rates.	HSP90 and CHIP complex form an interaction with ARF, which induces lysosomal degradation of ARF through binding to LAMP2A. The E3 ligase activity of CHIP is not required for formation of a tertiary complex and lysosomal degradation of ARF.	[78,79]
USP10	NSCLC	Positive correlation with ARF The combined expression of low USP10 and ARF worsen overall survival rates.	MYC increases the stability of ARF protein via induction of USP10, which is a deubiquitinase of ARF.	[75]
	Small intestine cancer	Positive correlation with ARF The combined expression of high USP10 and ARF are negatively correlated with vascular and lymphatic invasion. The combined expression of low USP10 and ARF worsen overall survival rates.	Several patients with intestinal adenocarcinoma contain aberrant hyper-methylations in the USP10 and ARF promoter regions with low expression of both proteins.	[119]
	Ovarian Cancer	Positive correlation with ARF The combined expression of low USP10 and ARF is displayed in cancer. The combined expression of low USP10 and ARF worsen overall survival rates.	High degree of methylation in USP10 and ARF CpG islands detected by methylation specific PCR analysis in ovarian cancer patients	[120]

(Related to ‘4. Post-translational regulation of ARF in human cancer’).

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
