# Peer review of "Post-Translational Regulation of ARF: Perspective in Cancer"

_biomolecules, 2020, doi:10.3390/biom10081143_

Round 1

Reviewer 1 Report

The review is very well written and provides a concise but comprehensive review on the molecular mechanisms that underlie the role of ARF is cancer.  

As the authors mention in line 153 of the manuscript, ARF protein is also regulated during embryogenesis. Although the review focuses mainly on the role of ARF in cancer, a brief comment to its roles in development could also be relevant. This is particularly interesting because such functions are also p53 independent, and unraveled a role for ARF that is quite unexpected. An example is the existence of the small mitochondrial ARF protein and its ability to correct developmental defects of ARF null mice (https://www.pnas.org/content/114/28/7420). This raises the hypothesis that at least some ARF functions in cancer and in development can be uncoupled and further underscores the complexity of ARF regulation in health and disease states.

However, this is just a minor suggestion.

Reviewer 2 Report

Interesting review summing up what we know about ARF in particular in cancer. Authors have made an extensive review of the transcriptional regulation and post-translational modifications that in the opinion of this reviewer covers the most important aspects to take into account. It is very informative and useful, and will make a valuable contribution to the field.

Reviewer 3 Report

Re: biomolecules-862338

Post-Translational Regulation of ARF: Perspective in Cancer

In this manuscript the authors describe the transcriptional and post-translational modifications of ARF, a tumor suppressor activating the ARF-MDM2-p53 axis. Furthermore, the authors discuss the regulatory mechanisms of ARF expression in different human cancers, highlighting the clinical significance of the molecule.

The manuscript is quite well written and faithfully recapitulates the recent relative bibliography. These issues need to be addressed.

Reviewer’s major remarks

  1. The abstract doesn’t really convey the content of the manuscript. It should mainly focus on ARF and its expression/modifications in cancer.
  2. Figure 1B, in the ARF-MDM2-p53 pathway, why do the authors show a “block” in p53 expression after MDM2 sequestration by ARF, since this leads to p53 activation?
  3. Tables need revision. Table I: Regulation by SMAD2/3, DMP1a, EZH2 is missing. Also, the second line in the column “expression status in cancer” is not very informative and should be re-written in order to better explain ARF expression in each type of cancer. The heading of the first column should be something other than mere “protein”, like “transcription factor” etc.  Table II: Same as above. Improvement of table II will add to the understanding of the relative chapter (ARF in cancer), which is quite complicated and difficult to follow.
  4. All abbreviations should be explained upon first reference.
  5. Several grammar mistakes throughout the text, eg lines 99-100, 108, 119, 67-70, 135-136, 143, 259 etc.

Round 2

Reviewer 3 Report

Re: biomolecules-862338

Post-Translational Regulation of ARF: Perspective in Cancer

The authors have addressed all the points raised and the manuscript is significantly improved. I have no other remarks.